# Developing and Regenerating Cofactors for Sustainable Enzymatic CO$_2$ Conversion

**Zhibo Zhang** [1,*] **, Xiangping Zhang** [2] **and Xiaoyan Ji** [1,*]

1 Energy Engineering, Department of Engineering Sciences and Mathematics, Lulea University of Technology, 97187 Lulea, Sweden

2 Beijing Key Laboratory of Ionic Liquids Clean Process, CAS Key Laboratory of Green Process and Engineering, State Key Laboratory of Multiphase Complex Systems, Institute of Process Engineering, Chinese Academy of Sciences, Beijing 100190, China; xpzhang@ipe.ac.cn

* Correspondence: Zhibo.zhang@ltu.se (Z.Z.); xiaoyan.ji@ltu.se (X.J.)

**Abstract:** Enzymatic CO$_2$ conversion offers a promising strategy for alleviating global warming and promoting renewable energy exploitation, while the high cost of cofactors is a bottleneck for large-scale applications. To address the challenge, cofactor regeneration is usually coupled with the enzymatic reaction. Meanwhile, artificial cofactors have been developed to further improve conversion efficiency and decrease cost. In this review, the methods, such as enzymatic, chemical, electrochemical, and photochemical catalysis, developed for cofactor regeneration, together with those developed artificial cofactors, were summarized and compared to offer a solution for large-scale enzymatic CO$_2$ conversion in a sustainable way.

**Keywords:** CO$_2$ conversion; enzyme; regeneration; natural cofactor; artificial cofactors





## 1. Introduction

Due to the rapid increase in energy demand and the excessive combustion of fossil fuels, the concentration of carbon dioxide (CO$_2$) in the atmosphere has been increasing at an alarming rate, which causes global warming, climate change, and severe environmental issues [1]. On the other hand, CO$_2$ has been identified as a cheap, abundant, and renewable carbon feedstock (750 billion tons of carbon in the atmosphere) for fuels and chemicals synthesis, such as formic acid, methanol, ethanol, etc. [2]. Therefore, the transformation of CO$_2$ into fuels and chemicals offers a win–win strategy for alleviating global warming and promoting renewable energy exploitation [3].

Intensive research has been conducted on CO$_2$ conversion over the past decades, including chemical, electrochemical, photochemical, and enzymatic conversions [4–6]. However, these methods exhibit their own drawbacks, impeding the large-scale application. For example, chemical conversion usually takes place under harsh conditions (high temperature and pressure), and a noble catalyst is needed, resulting in high energy usage and capital costs [1]. Photochemical conversion undergoes the instability of photosensitizer during the long-term light irradiation and the difficulty of downstream product separation [7]. Electrochemical conversion normally suffers from low selectivity with various products and low current density with poor efficiency [8]. Therefore, compared to the above methods, enzymatic conversion of CO$_2$ is a promising solution, as it can achieve CO$_2$ conversion with high selectivity and efficiency under mild and environmentally friendly conditions [9].

During the biotransformation of CO$_2$, formate dehydrogenase (FDH) serves as catalyst in the CO$_2$ reduction to formic acid, nearly without any by-products. By employing two additional enzymes, i.e., formaldehyde dehydrogenase (FaldDH) and alcohol dehydrogenase (ADH), CO$_2$ can be sequentially reduced to methanol [10]. In each step of this reduction process, nicotinamide adenine dinucleotide (NADH) acts as a sacrificial agent (substrate) to

provide hydrogen and electrons and finally is oxidized to $NAD^+$. Stoichiometrically, to produce one mole of methanol from $CO_2$, three moles of NADH are required [11]. In general, cofactor is much more expensive than the conversion product, and how to significantly reduce the cost of cofactor is essential to achieve an economically feasible and large-scale application. In addition, such a multi-enzymatic reaction is reversible, and the reverse reaction rate is enhanced during the progress of the reaction, resulting in suppressing product (methanol) generation.

To realize large-scale $CO_2$ biotransformation, the strategy for reducing $NAD^+$ to NADH (i.e., regeneration) was carried out and coupled with enzymatic conversion, which not only significantly decreases the cost but also alleviates the product inhibitory to improve production. In this review, various methods for NADH regeneration are summarized and discussed, including chemical, photochemical, electrochemical, and enzymatic regeneration. Furthermore, an artificial cofactor was also discussed and analyzed as a promising alternative to the expensive natural cofactor in order to achieve a sustainable enzymatic conversion of $CO_2$ and large-scale application.

## 2. Natural Cofactor Regeneration

### 2.1. Enzymatic Regeneration

Enzymatic cofactor regeneration is essential to achieve sustainable enzymatic $CO_2$ conversion. Typically, a second enzymatic reaction for NADH regeneration was integrated with an enzymatic $CO_2$ reaction (Figure 1), which can achieve $CO_2$ transformation in a sustainable way [12]. NADH regeneration not only provides enzymes with the reduced cofactor sustainably but also enhances $CO_2$ conversion, as by-products are removed and the NADH concentration is increased, driving the equilibrium forward and thus increasing $CO_2$ conversion [6]. Up to now, even though many enzymes are used for the enzymatic reduction of $NAD^+$ to NADH, only fewer are integrated with the enzymatic $CO_2$ conversion. It is probably because the enzymes for the NADH regeneration need to meet some requirements, such as high stability, compatibility with enzymatic $CO_2$ conversion systems, and high catalytic activity to compete with FDH for reducing $NAD^+$ to NADH. To date, glutamate dehydrogenase (GDH) and glucose dehydrogenase (GCDH) have been widely used in enzymatic $CO_2$ conversion, as they are stable, highly active, and commercially available. In addition, the yield of methanol was calculated based on the initial concentration of NADH, and the used equation is $Y_{methanol}$ (%) = $3*C_{methanol}/C_{NADH, initial}$. In the equation, $C_{NADH, initial}$ is the initial NADH concentration (mM), and $C_{methanol}$ was the methanol concentration.

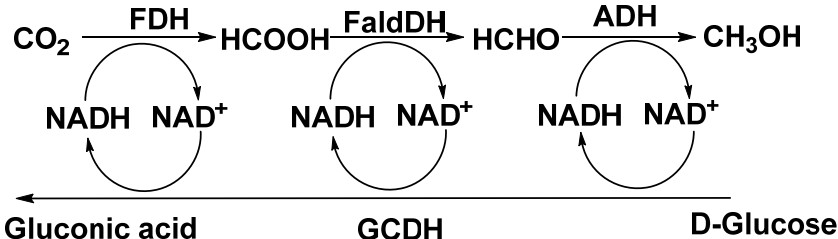

**Figure 1.** Multi-enzymatic conversion of $CO_2$ coupled with enzymatic regeneration of NADH by GCDH.

GDH is highly specific for the coenzyme $NAD^+$ that catalyzes the conversion of glutamate to $\alpha$-ketoglutarate and ammonia while reducing $NAD^+$ to NADH [13]. Importantly, this enzyme exhibits outstanding stability toward the variation of pH, chaotropic agents, organic solvents, and it is stable at high temperature, retaining 50% activity after 120 min incubation at 85 °C. Furthermore, the enzyme GDH is commercially available, inexpensive, and easy to manipulate and immobilize. Therefore, GDH has been recognized as one of the desirable NADH regeneration enzymes, together with FDH or two additional enzymes FaldDH and ADH, for $CO_2$ conversion in a sustainable way. In practice, GDH is typically

immobilized on novel materials, not only recycling enzymes to cut costs but also improving its catalytic performance in the microenvironment. In the following paragraphs of this section, immobilizing GDH on different biocompatible materials for sustainable cofactor regeneration integrating $CO_2$ reduction, such as hollow nanofibers, MOF (ZIF-8), magnetite nanoparticles, microparticles, etc., was reviewed.

Ji et al. [12] immobilized three enzymes, i.e., FDH, FaldDH, and ADH, in polyelectrolyte-doped hollow nanofibers for methanol production from $CO_2$, which was coupled with enzymes glutamate dehydrogenase and substrate glutamic acid for the reduction of $NAD^+$ to NADH. As a result, the yield of methanol reached up to 95.3% with enzymatic NADH regeneration. By contrast, without enzymatic NADH regeneration, the yield of methanol was only 33.1%. These results suggested that such a multi-enzymatic reaction arrives at equilibrium with a 33.1% methanol yield. After integrating with the NADH regeneration, the equilibrium was shifted, and the methanol yield was substantially promoted. Ren et al. [14] prepared a multi-enzyme nanoreactor by encapsulating FDH, cofactor (NADH), and glutamate dehydrogenases (GDH) into ZIF-8 for formic acid production from $CO_2$, which was further mixed with cationic polyelectrolyte that can capture cofactor NADH for immobilization. As a result, the nanoscale multienzyme reactor exhibited a superior capacity for the conversion of $CO_2$ to formate. Compared with the free multi-enzyme system, the formate yield was increased 4.6-fold by using the nanoscale multienzyme reactor. The nanoscale multienzyme reactor exhibited high stability and retained 50% of its original productivity after 8 cycles. Likewise, Caterina et al. [15] conducted multi-enzymatic conversion of $CO_2$ to methanol by using FDH, FalDH, and ADH, which was powered by $NAD^+/NADH$ and GDH as the coenzyme-regenerating system. All the enzymes were co-immobilized on the magnetite nanoparticles. As a result, the yield of methanol was increased 64-folds compared to the reaction without a regeneration system. This study demonstrated again the success of in situ NADH regeneration together with enzymatic $CO_2$ conversion, envisaging the possibility of using immobilized enzymes to perform the cascade $CO_2$-methanol reaction. Later, Bilal et al. [16] used a similar system for enzymatic methanol production from $CO_2$ with NADH regeneration by GDH but focused more on grafting cofactor NADH on the microparticles, which can be easily recovered and reused, effectively mediating the multistep reactions catalyzed by enzymes. In their work, FDH, FaldDH, ADH, GDH, and NADH were immobilized on the microparticles, and the reaction was performed by bubbling $CO_2$ with the collision between enzymes-immobilized particles to make sure that particles afforded sufficient interactions between the cofactor and enzymes. Over 80% of their original productivity was retained after 11 reusing cycles, with a cumulative methanol yield up to 127%, which was promising to the practical application. All these works demonstrated that GDH is an excellent candidate in the reduction of $NAD^+$ to NADH for sustainable enzymatic $CO_2$ conversion.

Glucose dehydrogenase (GCDH) is another useful catalyst for the conversion of beta-D-glucose to D-glucono-1,5-lactone while reducing $NAD^+$ to NADH. This enzyme also exhibits outstanding stability, which can tolerate high temperatures. In addition, its substrate glucose is compatible with most enzymes that have the advantage of integrating with enzymatic reaction. Moreover, the enzyme GCDH is commercially available, inexpensive, and easy to manipulate and immobilize, being a popular enzyme for reducing $NAD^+$ to NADH when coupling with enzymatic $CO_2$ conversion.

According to the membrane fouling mechanism, Zhang et al. [11] immobilized four enzymes, FDH, FaldDH, ADH, and GCDH, in the regenerated cellulose membrane for cofactor regeneration, integrating enzymatic $CO_2$ reduction. Among them, the first three enzymes were used for methanol production from $CO_2$, which was coupled with GCDH for reducing $NAD^+$ to NADH. The yield of methanol reached 73% in 30 min and then increased to 100% after coupling GCDH. The stability of immobilized enzymes was also proved, and the yield of methanol was maintained after 6 cycles, demonstrating the successful combination of enzymatic $CO_2$ conversion and NADH regeneration powered by GCDH and further revealing a facile enzyme immobilization method. Recently, Yu et al. [17]

reported the enzymatic formic acid production from $CO_2$ coupled with GCDH for the reduction of $NAD^+$ to NADH. The product formate concentration was accumulated quickly after initiating the reaction, but the reaction rate slowed down dramatically and without any increase in 10 h. After the verification via steady-state experiments, the product inhibition (formate) was the major reason for the rate decrease. Even though $NAD^+$ was removed and the NADH concentration was increased by GCDH, the product formic acid also needed to be removed from the system in order to increase product generation. Similarly, Marpani et al. [18] also performed a study to verify whether GCDH could meet the requirement for cofactor regeneration integrating enzymatic $CO_2$ reduction, and the reduction rate of $NAD^+$ to NADH was investigated and compared to the oxidation rate of NADH to $NAD^+$ by FDH. As a result, the reduction rate of $NAD^+$ to NADH catalyzed by GCDH (8.8 $\mu mol\ mg^{-1}$ $min^{-1}$) is faster than the oxidation of NADH to $NAD^+$ by ADH (4.7 $\mu mol\ mg^{-1}\ min^{-1}$). Therefore, GCDH could provide enough NADH amount for enzymatic conversion.

The enzymatic method for NADH regeneration has pros and cons in the reduction of $NAD^+$ to NADH. First, it provides an environmentally friendly solution for $NAD^+$ reduction, and it exhibits high selectivity and efficiency. Furthermore, the enzymatic reduction of $NAD^+$ is compatible with the enzymatic conversion of $CO_2$, achieving in situ NADH regeneration for continuous methanol production. In contrast, the enzymatic reduction of $NAD^+$ presents some disadvantages, such as instability of enzyme, high cost, and a complex system of product separation. Anyhow, the enzymatic method is a promising solution for cofactor regeneration integrating enzymatic $CO_2$ reduction.

### 2.2. Chemical Regeneration

The chemical method using a reducing reagent for the reduction of $NAD^+$ to NADH can be divided into two categories: inorganic salts and metal organometallic complex. Typically, inorganic salts, sodium dithionite ($Na_2S_2O_4$) [19,20], and sodium borohydride ($NaBH_4$) [21] is widely used, and sodium dithionite is a strong reducing reagent that can achieve hydrogen and electron transfer to cofactor $NAD^+$. As shown in Figure 2, the catalytic mechanism consists of four steps to reduce $NAD^+$ to NADH, as reported by Scriven et al. [19] Considering the low cost and high selectivity [22], sodium dithionite is a desirable candidate. However, sodium dithionite has a threshold concentration (0.1 M) to decrease the activity of enzymes, owing to the interaction of the thiol groups in the enzyme with dithionite. Moreover, since sodium dithionite is a small molecule, it is difficult to separate from the system, limiting the application in the enzymatic conversion. Likewise, sodium borohydride has a similar problem.

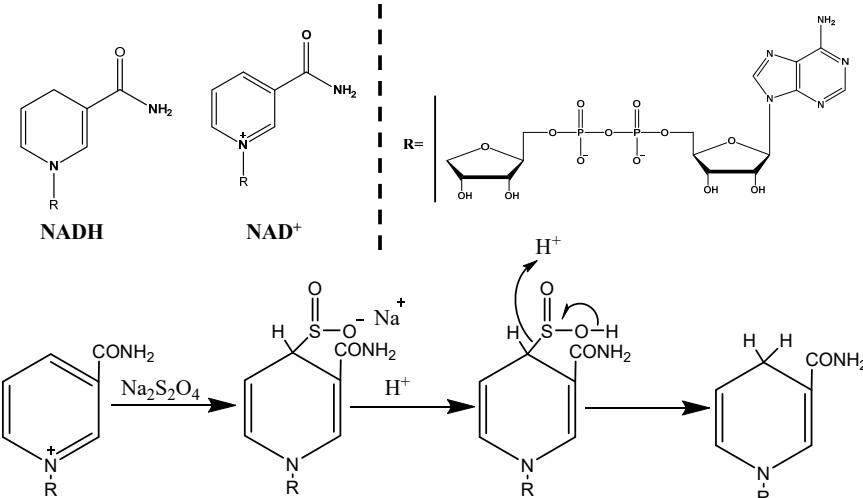

**Figure 2.** Mechanism of reducing $NAD^+$ to NADH by $Na_2S_2O_4$.

To achieve the NADH regeneration with high selectivity and efficiency, various organometallic complexes have been developed, such as Ir-based complex, Rh-based complex, and Fe-based porphyrins. Among them, the Rh-based catalysts $[Cp*Rh(bpy)]^{2+}$ ($Cp* = C_5Me_5$; bpy = 2,2′-bipyridine) are popular, owing to their high stability and selectivity over a broad range of pH and temperature as well as the compatibility with the enzymatic system. Such catalysts have stable reduced forms and oxidized states in the homogeneous reduction system, which potentially achieve interconversion by the external driving force. In the enzymatic conversion of $CO_2$ to formic acid, the cofactor NADH is oxidized to $NAD^+$ by providing two electrons and one hydrogen to $CO_2$ and forming formate. The $Rh^I$-based catalysts can efficiently reduce $NAD^+$ to NADH while oxidizing $Rh^I$ to $Rh^{III}$, indicating that two electrons donated from $Rh^I$ are transferred to $NAD^+$ for generating NADH. As shown in Figure 3, $CO_2$ can be converted to formic acid or methanol while oxidizing NADH to $NAD^+$, and then, the Rh-based catalysts are used to regenerate NADH by providing electrons. However, the Rh-based catalyst cannot be recovered after reducing $NAD^+$ to NADH while $Rh^I$ is oxidized to $Rh^{III}$, which is unsustainable and limits its application. Therefore, the recovery of Rh-based catalysts has been developed by combining the photochemical and electrochemical methods.

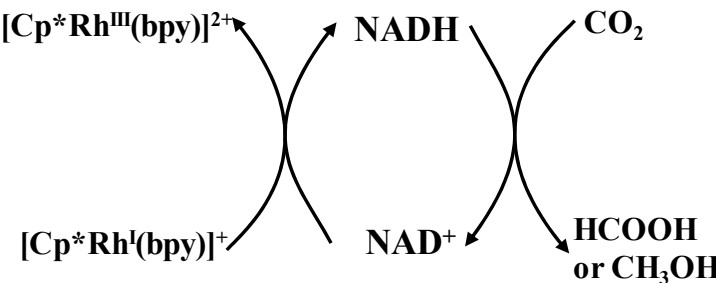

**Figure 3.** Enzymatic conversion of $CO_2$ coupled with NADH regeneration by Rh complex.

### 2.3. Photocatalytic Regeneration

To achieve NADH regeneration coupling with $CO_2$ conversion using the photocatalytic method, natural photosynthesis would be an ideal model. In natural photosynthesis, NADH is regenerated via multistep electron transfer, which is known as the Calvin cycle, where chlorophyll works as a photosensitizer, playing a key role in light-harvesting systems [23]. To establish an NADH regeneration system driven by photosynthesis, photocatalytic NADH regeneration should contain a photosensitizer, electron mediator, and electron donor [24]. Herein, the electron mediators can be the Rh-based catalysts, as described in the chemical regeneration of NADH, and the electron donors can be water or electron-rich chemicals (triethanolamine). The photosensitizer is used for recovering the electron mediators. The photocatalytic regeneration of NADH can be coupled with enzymatic $CO_2$ conversion, where both inorganic and organic photosensitizers are introduced for regenerating NADH.

#### 2.3.1. Inorganic Photosensitizer

$TiO_2$ has been widely used as photocatalysts in environmental and energy applications, owing to its high photoactivity, stability, safety, and low cost [25]. However, its band energy gap, 3.2 eV, limits the absorption of solar radiation, which accounts for only 4% of the solar spectrum [26]. To improve photoactivity, many efforts have been devoted to modifying $TiO_2$ with quantum dots or nanoparticles to expand the energy bandgap with more negative conduction. To date, using doped carbon, boron, nitrogen, phosphorous, and CdS on $TiO_2$ as photosensitizers and combining with the Rh-based complex have been extensively investigated. These quantum dots or nanoparticles can be tunned with size and surface-to-volume ratio, and typically, smaller quantum dots in size enhance photoactivity [27]. As shown in Figure 4, under irradiation, the photo-excited electrons will transfer from the

electron donor (TEOA) to the Rh-based complex with the help of a photosensitizer, [28] and thus, Rh$^{III}$ is reduced to Rh$^{I}$, and the reduction of NAD$^+$ to NADH catalyzed by the Rh$^{I}$ complex can be achieved.

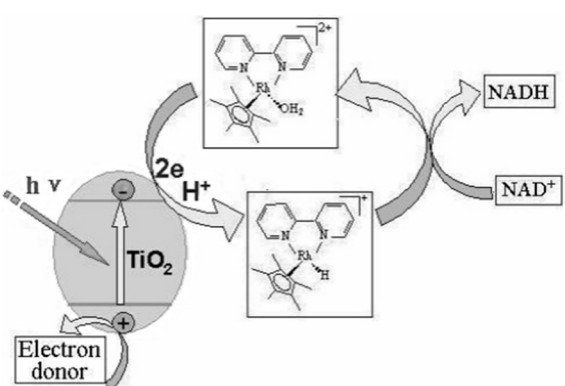

**Figure 4.** Photosystem for the NADH regeneration by using the TiO$_2$ and Rh complex.

Non-metal elements, such as carbon, boron, nitrogen, and phosphorus, have been firstly attempted to dope on TiO$_2$ to examine their photocatalytic performance. For example, Jiang et al. [28] prepared carbon-contained TiO$_2$ as a photosensitizer and used it for the photochemical regeneration of NADH. As a result, the carbon-doped TiO$_2$ causes a red-shift of the absorption edge and thus enhances the absorption of visible light. Under the catalysis of the Rh-based complex, the carbon-doped TiO$_2$ exhibits higher activity and selectivity compared to the pristine TiO$_2$. In addition, the effects of different conditions, electron donors, pH values, and Rh complex concentrations were evaluated. Similarly, boron, nitrogen, and phosphorus were doped on TiO$_2$ for photoactivity testing [29–31], and again, the doped TiO$_2$ exhibits higher photoactivity compared to the pristine TiO$_2$. However, as the first generation of heterogeneous inorganic photosensitizers, the TiO$_2$-based photosensitizers generally show a much lower activity (TOF: 0.031 h$^{-1}$) compared to the organic photosensitizers.

Metal-based quantum dots have been developed for the photocatalytic regeneration of NADH. Dong et al. [32] immobilized CdTe, CdSe, and CdS on TiO$_2$ to evaluate their photocatalytic performance on the NADH regeneration (Figure 5). To make a comparison, the pristine nanoparticles of CdS, CdSe, and CdTe were also separately evaluated on their photocatalytic activity, evidencing no reduction of NAD$^+$ to NADH at all. After immobilizing these nanoparticles on TiO$_2$, these modified TiO$_2$ nanoparticles have a high capability to drive the in situ photochemical regeneration of NADH under visible light (420 nm). The CdTe-modified photosensitizer presented the best photocatalytic performance (TOF: 0.54 h$^{-1}$), which is much higher than the non-metal quantum dots. As expected, the yield of NADH is higher with smaller-sized nanoparticles, which may be attributed to their enhanced photo efficiency and increased number of active sites.

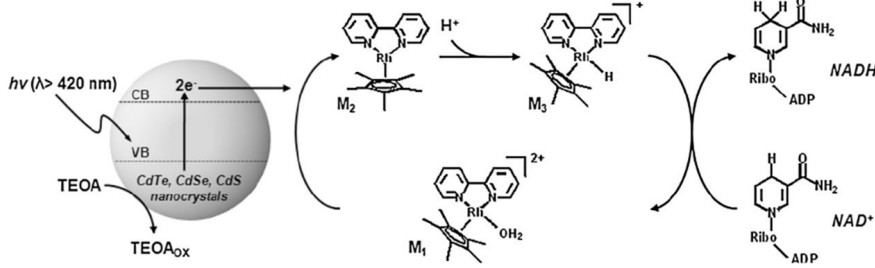

**Figure 5.** Schematic diagram of photochemical regeneration of cofactor by using CdTe, CdSe, and CdS nanocrystals.

### 2.3.2. Organic Photosensitizer

To develop a more efficient photocatalytic system for $CO_2$ conversion, highly efficient natural photosynthesis has attracted much attention and been used as the reference for mimicking. As shown in Figure 6, natural photosynthesis has a cycle of NADH regeneration integrating with the enzymatic conversion of $CO_2$ to sugar, which is known as the Calvin cycle [33]. To address the challenges of developing efficient and stable photosensitizers, inspired by the main composition and structure of chlorophyll, light-harvesting porphyrins attracted great interest (Figure 6). Jae et al. [33] immobilized tetra (phyroxyphenyl) porphyrin (THPP) on nanotubes, which not only improve the photosensitizer's stability but also promote excited-electron transfer to the Rh complex, realizing artificial photosynthesis with NADH regeneration and enzymatic conversion.

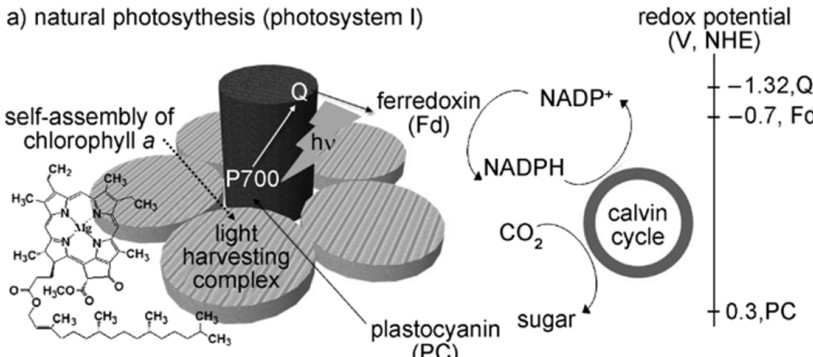

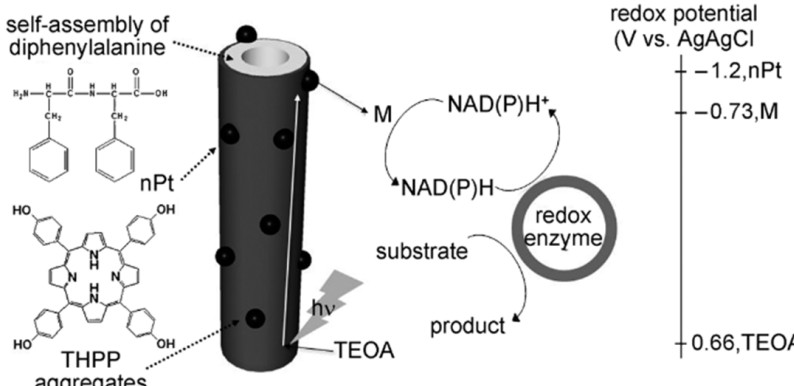

**Figure 6.** (**a**) Natural photosynthesis; (**b**) biomimetic photosynthesis.

Following the work by Kim et al. [34], the authors focused on functioning porphyrins with sulfonate or carboxyl groups and modified an active site in the center of the porphyrin ring to evaluate the optimal photosensitizers. Six porphyrin-containing light-harvesting molecules (Figure 7), which contain different metals (Zn or Mn) and functional groups (sulfonate or carboxyl groups), were studied. The Zn-tetrakis (4-carboxyphenyl) porphyrin is the most efficient one with a turnover frequency of 0.46 $h^{-1}$, which is much higher than other photosensitizers, such as CdSe (0.168 $h^{-1}$), CdS (0.120 $h^{-1}$), P-doped $TiO_2$ (0.003 $h^{-1}$), and $W_2Fe_4Ta_2O_{17}$ (0.002 $h^{-1}$). This work demonstrated that a functionalized porphyrins ring could efficiently facilitate the generation of photo-excited electrons, and the metal zinc in the center with more electronegativity would enhance the electron transfer from the Rh complex.

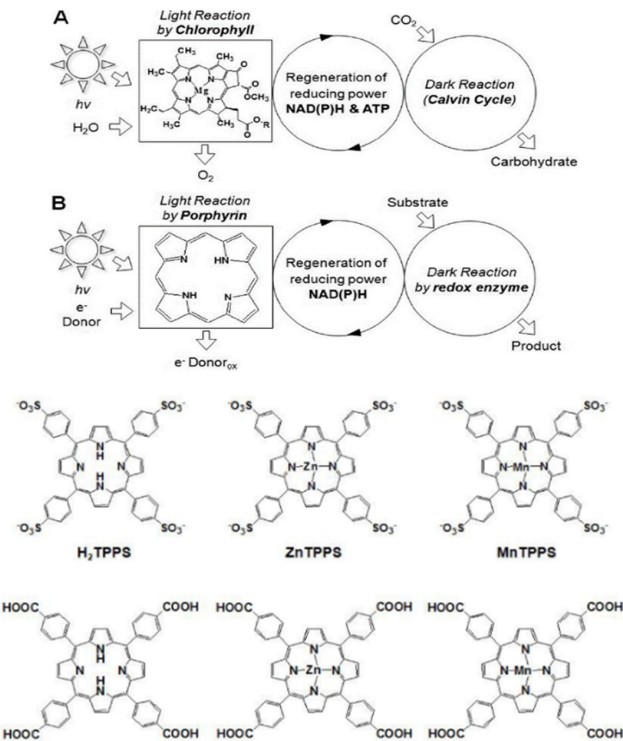

**Figure 7.** (**A**) Natural photosynthesis; (**B**) biomimetic photosynthesis using photosensitizers; chemical structure of photosensitizer in the below.

Likewise, Zhang et al. [35] establish artificial photosynthesis by modifying the porphyrin with ionic form (Figure 8), which is more hydrophilic and conductive that can significantly enhance the reduction of $NAD^+$ to NADH. By integrating with a multi-enzymatic reaction, methanol was produced at the rate of 0.16 mM·h$^{-1}$, which is much higher than the most reported results. Therefore, Zn-porphyrin is a promising light-harvesting molecule for the biocatalyzed artificial photosynthesis.

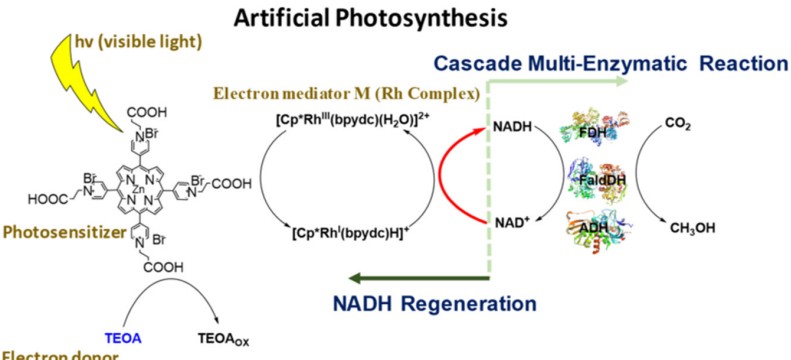

**Figure 8.** Ionic porphyrin as the photosensitizer is used for the photocatalytic reduction of $NAD^+$ to NADH providing the cofactor in a reduced form for the enzymatic synthesis of methanol.

Although organic photosensitizers showed high efficiency, most of these photosensitizers suffer from structural instability and difficulty in recycling. Therefore, an immobilizing porphyrin-based photosensitizer was carried out. Rajest et al. [36] grafted porphyrin on the graphene sheet (CCGCMAQSP), as shown in Figure 9 for integrating enzymatic $CO_2$ conversion, and established artificial photosynthesis for producing formic acid [37]. Compared to the photosensitizer $W_2Fe_4Ta_2O_{17}$, the yield of NADH using CCGCMAQSP is 5 times higher in two hours, while the yield of formic acid is more than 7 times higher. Therefore, the porphyrin-immobilized graphene shows higher photoactivity and is more compati-

ble compared to the inorganic photosensitizer $W_2Fe_4Ta_2O_{17}$, owing to the high stability, conductivity, and compatibility of support graphene. Later, the same group performed a multi-enzymatic reaction using the same strategy for the methanol production from $CO_2$, demonstrating again the success of this artificial photosynthesis system [38]. According to the review investigation, organic photosensitizers exhibit three to 100 times better catalytic activity than inorganic semiconductors, whose synthetic processes are generally more complicated.

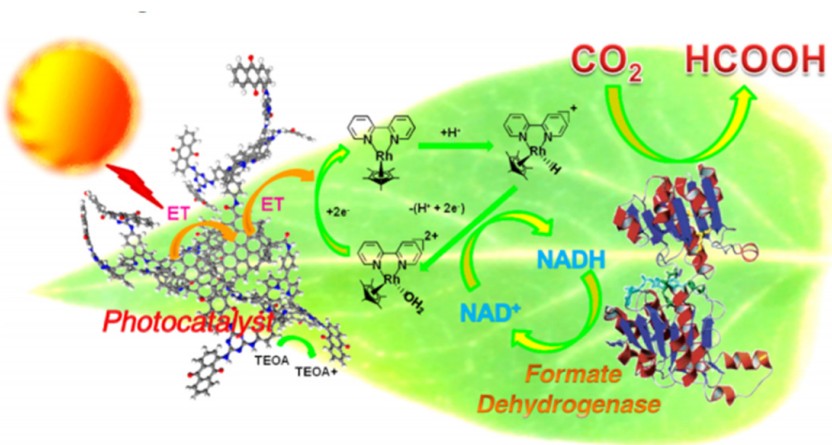

**Figure 9.** An artificial photosynthetic system for producing formate from $CO_2$ with MAQSP-CCG as photocatalyst.

### 2.4. Electrochemical Regeneration

Electrochemical NADH regeneration is a promising solution and can be potentially used in a large-scale application, as electrocatalysis can provide a stable current density at a low cost. Moreover, it is more "clean" compared to other methods without by-product separation. The electrochemical regeneration of NADH can be categorized as direct and indirect regeneration. The direct reduction can be accomplished by receiving two electrons and one proton. The mechanism can be briefly summarized in three steps [26,39]. As shown in Figure 10, under the potential from the cathode, $NAD^+$ firstly receives one electron to form radical species $NAD^{+\cdot}$. Then, the radical $NAD^{+\cdot}$ will be protonated and finally receive the second electron to generation NADH (1,4-NADH) [40]. The electrochemical reduction of $NAD^+$ also suffers from low selectivity, and by-products ($NAD_2$ and 1,6-NADH) are usually formed under the potential of higher $-1.6$ V in the second step [41]. To improve the selectivity, various metal-based electrodes were prepared to enhance the kinetics of the second reaction step, and thus, the potential can be decreased and the selectivity is improved [26].

**Figure 10.** Reaction pathways in electrochemical reduction of $NAD(P)^+$ to $NAD(P)H$.

To date, the metal-based nanostructured electrodes, Au, Pt, Ti, Co, Cd, Ir, Ru, and Ni, have been used to reduce $NAD^+$. Omanovic et al. [42] tested the Au electrode with different potentials. At the potential $-1$ V, the yield of NADH can reach 75% but with a low reduction rate. At the potential $-1.1$ V, the reduction rate can be markedly improved but suffers from a low yield (28%). By further modifying the electrode with Pt, the yield of NADH is increased from 28 to 63% at the potential of $-1.1$ V, owing to the kinetic enhancement of the second step reaction and speeding up the protonation process by Pt. Later, the same group prepared a new electrode by depositing Pt and Ni nanoparticles on the glass carbon, achieving the yield of NADH almost 100% at more negative potentials of $-1.5$ V, which is due to the protonation enhancement by Pt at the second step [43]. Similarly, more metal-based electrodes, Ti, Co, Cd, Ir, Ru, and Ni, were developed, and the Ti electrode presented the best performance with a yield of 96%, attributing to the strength of the Ti-hydrogen bond [44]. Minteer et al. developed cobaltocene-modified poly-(allylamine) redox polymer, and the yield reaches 97–100%, faradaic efficiencies 78–99%, and turnover frequencies between 2091 and 3680 $h^{-1}$ over a range of temperatures spanning 20 to 60 °C [45]. In addition to the metal-based electrode, some novel materials were also used for electrochemical NADH regeneration. Chan et al. [46] fabricated the non-conductive and electrically conductive silica gels, and they found that by using conductive silica gels, the turnover number of $NAD^+$ reduction was three-fold higher than that of the non-conductive silica gels. Furthermore, bare multi-walled carbon nanotubes were used as a cathode for electrochemically reducing $NAD^+$, and the yield of NADH reached 98% at a higher cathodic potential ($-2.30$ V) [47].

Compared to the direct regeneration of NADH, the indirect regeneration typically exhibits high selectivity. The characteristic of the indirect method employs an electron mediator, which acts as an electron carrier and achieves two electrons and one proton transfer in one step. Similar to the photochemical regeneration of $NAD^+$, the Rh complex has also attracted attention in electrochemical regeneration, owing to its high selectivity for reducing $NAD^+$ and stability of different valences of the Rh complex. Compared to the photochemical regeneration, electrochemical regeneration is much more "clean" without using photosensitizer and electron donors, which is beneficial to reduce cost and separate product in the downstream operation. For instance, Kim et al. [48] used $[Cp^*Rh(bpy)Cl]^+$ complex as a mediator and copper electrode for electrochemically reducing $NAD^+$ and applied it in the enzymatic conversion of $CO_2$ to formic acid. The applied potential was less negative ($<-0.9$ V vs. SCE) value, avoiding the generation of by-products $NAD_2$ and 1, 6-NADH (1.6 V). The selectivity of $NAD^+$ reduction to NADH is almost 100%, and the established system is much more simple and efficient compared to the photochemical method. Following this work, the Rh complex was grafted on the electrode to facilitate the product separation and recycling the Rh complex. Zhang et al. [49] developed a method to graft the Rh complex on a porous carbon electrode. As shown in Figure 11, the azido-functionalized electrodes were firstly prepared by the electrochemical reduction of 4-azidophenyl diazonium cations. Second, the bipyridine ligands were attached to the azido group via a Huisgen cycloaddition reaction. Last, the $(RhCp^*Cl_2)_2$ dimer was coordinated to the bipyridine ligands, and thus, the Rh complex-grafted electrode was prepared with the covalent grafting for the electrochemical reduction of NAD to NADH. As a result, the Rh complex-grafted electrode presented a high Faradaic efficiency of 87% and high stability with more than 90 h, making this system promising for the application to the electroenzymatic synthesis. Later, a more easily grafting method was developed [41] by replacing the bipyridine ligand with 2,2′-bipyridyl-5,5′-dicarboxylic acid. By alkalifying the electrode with the generation of the hydroxide group, the Rh complex with the carboxylic group was directly covalently immobilized to the electrode, achieving almost 100% selectivity.

**Figure 11.** Synthetic route followed for the functionalization of a carbon electrode surface with Rh complex.

Herein, as described above, there are some advantages and disadvantages of the four strategies for natural cofactor regeneration, which are briefly summarized in Table 1. To achieve the large-scale or industrial application of enzymatic conversion of $CO_2$, cofactor regeneration is the major bottleneck, and some drawbacks need to be solved by developing more advanced technology. The enzymatic method is restricted by the high cost of enzyme and coenzyme. The chemical method is limited by salts concentration and unsustainability. The photochemical method is complex, including photosensitizer, electron mediator, electron donor, enzymes, and coenzymes. Such a complex system would usually create the problem of product separation in downstream. Furthermore, a photosensitizer usually suffers from poor stabilization and easy decomposition in the long run. Since enzyme activity is sensitive to temperature, the system, under light irradiation in the long run, suffers from temperature increases, leading to a decrease in enzyme activity. The electrochemical method is easier to achieve, being a promising solution.

**Table 1.** Advantages and disadvantages of the four strategies for coenzyme regeneration.

| NADH Regeneration Solution | Advantages | Disadvantages |
|---|---|---|
| Enzymatic method | Environmentally friendly, high efficiency, high selectivity | High cost of enzymes and coenzymes. By-product separation, instability of enzyme |
| Chemical method | Low cost | By-products separation, unfriendly environment, unfriendly to enzyme, low efficiency, low selectivity, unsustainable. |
| Photochemical method | Low cost, environmentally friendly, the efficiency of energy utilization | Unstable photocurrent, unstable photosensitizer, byproduct separation |
| Electrochemical method | Low cost, environmentally friendly, high efficiency | Instability of electrocatalyst, low selectivity. |

## 3. Artificial Cofactor Development and Regeneration

Artificial cofactors recently received much attention, and they are more cheap and efficient in the enzymatic conversion of $CO_2$ compared to the natural cofactor. When using a natural cofactor, the enzymatic conversion of $CO_2$ is reversible as shown in Figure 12 (Equation 1), and the reverse reaction rate is 30 times higher than the $CO_2$ conversion rate, leading to the low conversion of $CO_2$ [50]. Furthermore, the natural cofactor is unstable and easily generates isomers such as $NAD_2$ dimer and inactive 1, 6-NADH in the recovery. By using an artificial cofactor, the reverse reaction can be inhibited as shown in Figure 12

(Equation 2), resulting in the improvement of $CO_2$ conversion. The artificial cofactor is an ionic salt, which is stable and gives possible long-term operation and recycling.

$$CO_2 + NADH \overset{FDH}{\rightleftharpoons} HCOO^- + NAD^+ \qquad (1)$$

$$CO_2 + 2BP^{+\cdot} + H^+ \overset{FDH}{\rightleftharpoons} HCOO^- + 2BP^{2+} \qquad (2)$$

**Figure 12.** (**1**) Enzymatic conversion of $CO_2$ with natural cofactor; (**2**) enzymatic conversion of $CO_2$ with artificial cofactor.

Mimicking the natural cofactor, an artificial cofactor should play as an electron carrier supplying to FDH for $CO_2$ conversion. Bipyridine salts can be desirable candidates, as they can achieve electron acceptance and release by the external driving force. Similar to the natural cofactor, the artificial cofactor has both oxidized and reduced forms, and the reduced form could be used in the enzymatic conversion of $CO_2$ by providing electrons. The artificial cofactors that have been developed are summarized in Figure 13. All of them are used in the enzymatic conversion of $CO_2$ combined with chemical, photochemical, or electrochemical regeneration.

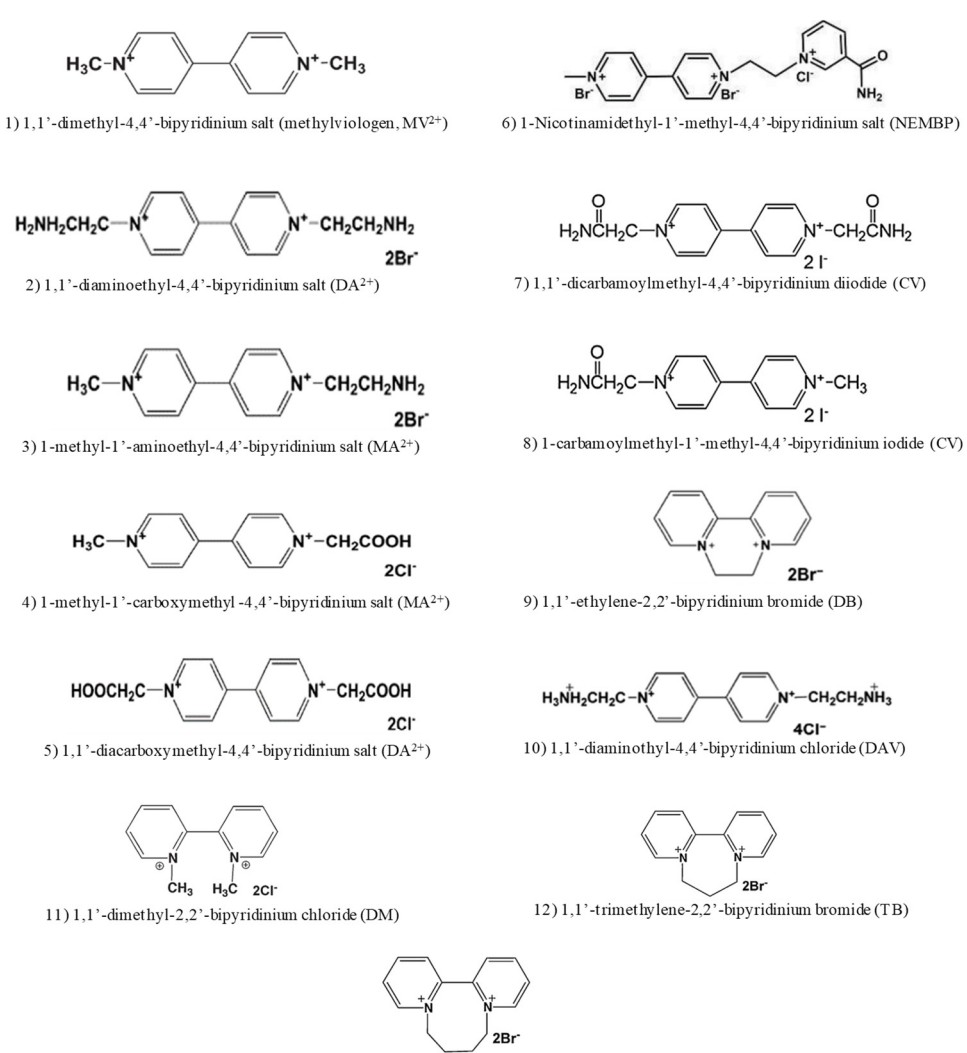

**Figure 13.** Chemical structure of (1–5) from Ref. [51]; (6) from Ref. [52]; (7,8) from Ref. [53]; (9,10) from Ref. [54]; (11–13) from Ref. [55].

Methyl-functioned bipyridine is a well-known artificial cofactor (methylviologen (1) in Figure 13) for FDH catalyzing $CO_2$ conversion. Amao et al. [56] investigated the kinetic properties of the enzymatic conversion of $CO_2$ with formic acid as the target product by using methylviologen and chemical reductant sodium dithionite for its regeneration. The results demonstrated that the reverse reaction (formic acid $\rightarrow CO_2$) was suppressed. Later, the same group developed a series of bipyridine-based derivates as cofactor ((1)–(5) in Figure 13), and the bipyridine with two amino groups, 1,1′-diaminoethyl-4,4′-bipyridinium salt, shows best catalytic efficiency, which is 560 times greater than NADH [51].

In the chemical regeneration of an artificial cofactor, sodium dithionite is used, which is a strong reductant reagent. A threshold concentration (0.1 M) will be deleterious to enzymes owing to the interaction of thiol groups in enzyme with dithionite. Similar to the natural cofactor regeneration, the photochemical regeneration received attention. Tomoya et al. [57] used $TiO_2$ as the photocatalyst, and $MV^{2+}$ was used as an electron mediator and artificial cofactor for the enzymatic conversion of $CO_2$. The authors successfully developed an effective light-driven hybrid system for reducing $CO_2$ to formic acid. Compared to the photochemical regeneration of NADH, the system for artificial cofactor regeneration is more simple and efficient.

Likewise, Takumi et al. [58] developed $CdS/CuInS_2$ electrodes for the photoelectrochemical regeneration of artificial cofactor integrating with enzymatic reaction. As shown in Figure 14, the established hybrid system provides a highly stable photocurrent with approximately 100% faradaic efficiency for the reduction of $MV^{2+}$ to $MV^{+\cdot}$. In addition, the author proposed the mechanism of $CO_2$ reduction with two molecules of $MV^{+\cdot}$. Combining the skeleton of artificial cofactor (bipyridine) and natural cofactor (pyridine ring), a new artificial cofactor was synthesized, as shown in Figure 13 (number 6), achieving electron and hydrogen transfer [52]. By using water-soluble zinc tetraphneylporphyrin tetrasulfonate as the photosensitizer, the turnover numbers for $CO_2$ reduction to formate reached 1.53 $h^{-1}$.

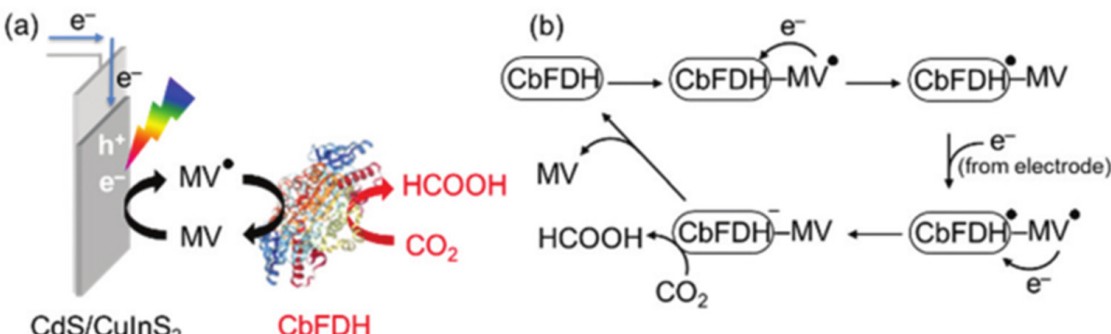

**Figure 14.** (**a**) Photoelectrochemical $CO_2$ reduction to formate over a hybrid system of $CdS/CuInS_2$ photocathode and FDH, and (**b**) $CO_2$ reduction to formate with FDH and two molecules of $MV^{+\cdot}$.

The photochemical regeneration system consists of a photosensitizer, electron donor, and artificial cofactor, which brings difficulty to the separation in the downstream operation. Therefore, electrochemical regeneration was investigated. As shown in Figure 15, Zhang et al. [59] developed a series of ionic bipyridine-based salts combined with electrochemical regeneration method for the enzymatic conversion of $CO_2$ in a sustainable way. As a result, the artificial cofactor 1,1′-diaminoethyl-4,4′-bipyridinium bromine exhibits the highest catalytic efficiency ($k_{cat}/K_m$), which is 536 times higher than that of NADH. The desirable performance was further explained by molecular dynamics simulations, demonstrating that the cofactor with the amino groups had the highest affinity for $CO_2$.

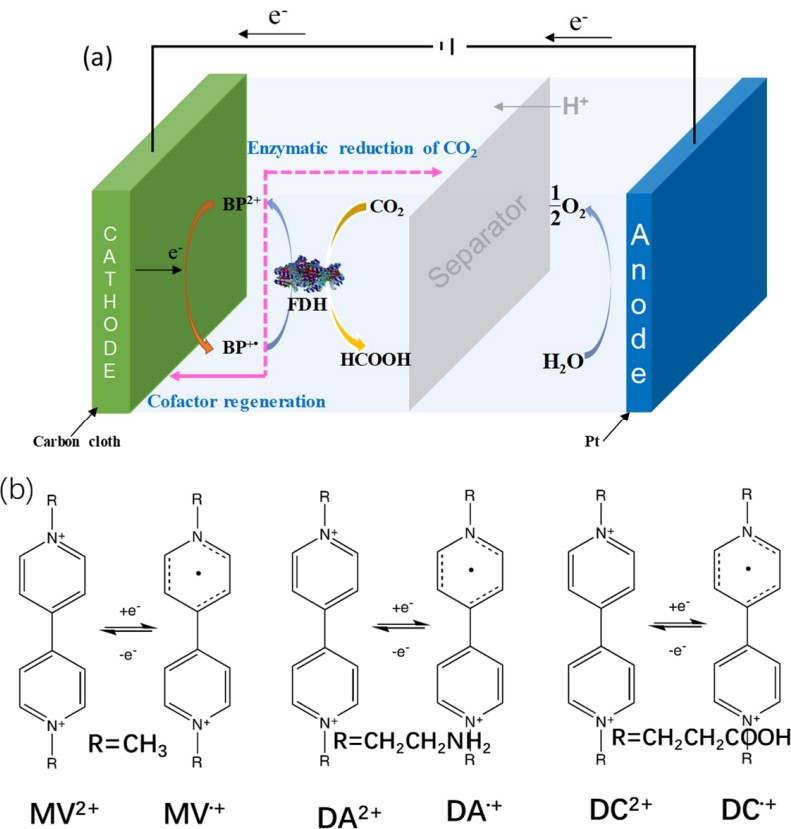

**Figure 15.** (**a**) The bioelectrocatalytic reduction of $CO_2$ to formic acid, and (**b**) chemical structures of the studied bipyridinium-based artificial redox cofactors.

Toward the practical application, Buddhinie et al. [60] designed a novel three-compartment cell separated by ion-exchange membranes to continuously electro-enzymatically produce formate using artificial cofactor and in situ separate the methyl viologen radical cation and formate (Figure 16). Formate yields as high as $97 \pm 1\%$ could be realized by avoiding the adventitious reoxidation of the methyl viologen radical cation by molecular oxygen. These results demonstrated that the artificial cofactor could potentially substitute the natural cofactor, and they are more efficient, stable, and cheap. Combining the electrochemical regeneration, the large-scale application of enzymatic $CO_2$ conversion can be realized in a sustainable way.

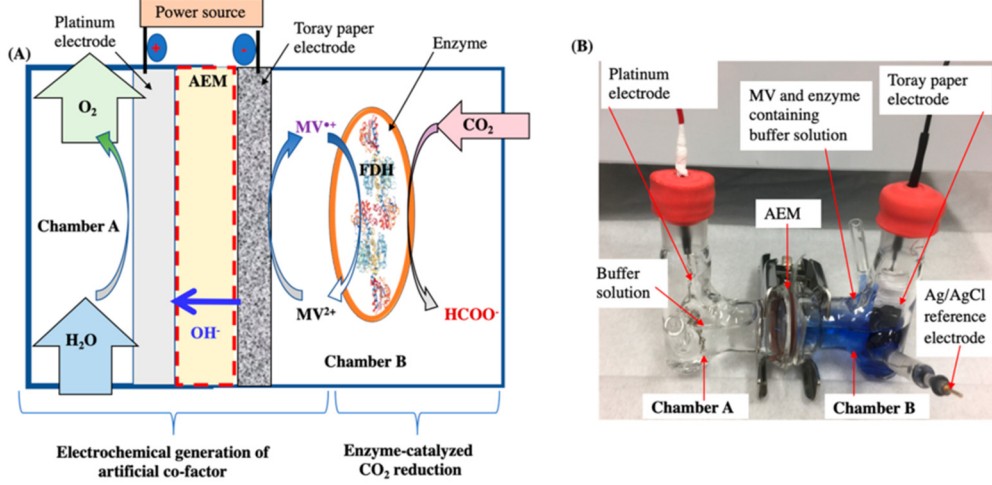

**Figure 16.** Electrochemical formate generation: (**A**) reactor configuration, (**B**) experimental setup.

With the development of artificial cofactors, it is promising to substitute natural cofactors, and the artificial cofactors exhibit more stability, efficiency, and low price, being promising to promote and realize the large-scale application. In addition, artificial cofactors can be regenerated by chemical, photochemical, and electrochemical methods, and the electrochemical regeneration is also a priority. Up to now, the enzyme cost and formic acid separation are two more challenges for the large-scale application of enzymatic $CO_2$ conversion. We envisioned that the immobilization of enzymes with a long lifetime would significantly cut down the cost, and the development of an advanced membrane would achieve a low concentration of formic acid separation continuously, making it possible to achieve the large-scale application of enzymatic $CO_2$ conversion.

## 4. Conclusions

As cofactor regeneration is the major bottleneck of enzymatic conversion of $CO_2$, this review summarized four strategies for natural cofactor regeneration, combining the enzymatic conversion of $CO_2$, which can establish the efficient enzymatic conversion of the $CO_2$ system in a sustainable way. In addition, artificial cofactors open another door in enzymatic conversion, which can inhibit the oxidation of formate and enhance $CO_2$ conversion. The current reported artificial cofactors were summarized, and their regeneration by chemical, photochemical, and electrochemical methods was discussed. The artificial cofactor is more stable, and the electrochemical regeneration of artificial cofactor is more "clean" and efficient in our opinion, being promising to realize the large-scale enzymatic conversion of $CO_2$.

**Author Contributions:** Conceptualization by Z.Z. and X.J.; Writing and main draft preparation by Z.Z. and X.J.; Figures drawing by Z.Z.; Review and editing by Z.Z., X.Z. and X.J.; Visualization and supervision by X.J. The Project was funded by X.J. All authors have read and agreed to the published version of the manuscript.

**Funding:** This research was funded by the Kempe Foundation in Sweden.

**Institutional Review Board Statement:** Not applicable.

**Informed Consent Statement:** Not applicable.

**Data Availability Statement:** Not applicable.

**Conflicts of Interest:** The authors declare no conflict of interest.

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
