# Peer review of "Developing and Regenerating Cofactors for Sustainable Enzymatic CO2 Conversion"

_processes, doi:10.3390/pr10020230_

Round 1
Reviewer 1 Report
Comments are reported in the attached file.

Author Response
Reply: Thank the reviewer for your valuable comments and suggestions for improving our manuscript. The yield of methanol has been clarified and added/revised in the revised manuscript. When the yield was more than 100%, it indicated that the initial concentration of NADH was completely consumed and then get regeneration.
Page: 2, lines 77-80
“Besides, the yield of methanol was calculated based on the initial concentration of NADH, and the used equation is: Ymethanol (%) = 3*Cmethanol/CNADH, initial. In the equation, CNADH, initial is the initial NADH concentration (mM), and Cmethanol was the methanol concentration.”
Reviewer 2 Report
The presented review is devoted to the technologies of enzymatic conversion of CO2, which are very important for reducing global warming on the planet and using renewable energy sources. The main attention is paid to the consideration of the chemical steps of enzymatic reactions with the presentation of detailed reaction schemes. The enzymatic reactions used take place with the participation of cofactors, which significantly complicates and increases the cost of the technology. Therefore, the enzymatic, chemical and photocatalytic methods of cofactor regeneration or using artificial cofactors are in the focus of this review. The volume and the amount of cited literature are quite sufficient. Without any doubt, the review will be of interest for the readers of Processes.
I think that the article can be improved taking into account the following comments and suggestions:
- The review contains a large number of chemical schemes and this is good, but since the processes are multi-enzyme and the substrates and cofactors are quite complex chemical compounds, the drawings look overwhelmed. At the same time, the authors presented all the schemes in approximately the same space, and some structural formulas are so reduced that it is difficult to make out what is shown there. This applies at least to Figures 6, 7, 12, 14. More space should be given for these figures. Alternatively, authors could enter the designation of chemical compounds by numbers, for example, and present the chemical formulas in the text or in the figure captions.
- Does it make sense to show the spatial structures of enzymes in the figures? If the location of the functional groups or the secondary structure is important, then it is necessary to provide links from which these data are taken. If they are given just like that, then you can omit them and thus simplify the figures.
- It makes no sense to designate two reactions in a special way, like equations 1 and 2 (lines 389-390). Authors should designate them as a figure.
- Table 1: “Disadvantages for Electrochemical method are stability of electrocatalyst, selectivity”. Maybe there should be instability and low selectivity.
Author Response
- The review contains a large number of chemical schemes and this is good, but since the processes are multi-enzyme and the substrates and cofactors are quite complex chemical compounds, the drawings look overwhelmed. At the same time, the authors presented all the schemes in approximately the same space, and some structural formulas are so reduced that it is difficult to make out what is shown there. This applies at least to Figures 6, 7, 12, 14. More space should be given for these figures. Alternatively, authors could enter the designation of chemical compounds by numbers, for example, and present the chemical formulas in the text or in the figure captions.
Reply: Thank the reviewer for the comments and suggestions. Figures 6, 7, 12, and 14 have been enlarged properly to make sure that chemical structures can be identified.
Chemical compounds in Figure 12 have been named with initials under the structure. Take the 1,1’-dimethyl-4,4’-bipyridinium salt as an example, it was initialized with MV2+, and their initials were also used in the text.
- Does it make sense to show the spatial structures of enzymes in the figures? If the location of the functional groups or the secondary structure is important, then it is necessary to provide links from which these data are taken. If they are given just like that, then you can omit them and thus simplify the figures.
Reply: Thank the reviewer for the comments. The figures were directly taken from the published articles, which are in general difficult to replot, and thus the original figures were represented in this review with citations.
- It makes no sense to designate two reactions in a special way, like equations 1 and 2 (lines 389-390). Authors should designate them as a figure.
Reply: Thank the reviewer for the comment. According to the suggestions from the reviewer, these two equations have been revised and designated as a figure in the revised manuscript.
- Table 1: “Disadvantages for Electrochemical method are stability of electrocatalyst, selectivity”. Maybe there should be instability and low selectivity
Reply: Thank the reviewer. The expression have been corrected as “instability of electrocatalyst, low selectivity” in the revised manuscript.
Reviewer 3 Report
The authors presented an interesting review of developing and regenerating cofactors for sustainable enzymatic CO2 conversion.
There are practically no comments. Confuses the yield of methanol - 127% (Line 122). Why such a value? Regarding what? This needs to be explained.
Author Response
Reply: Thank the reviewer for your valuable comments and suggestions for improving our manuscript. The yield of methanol has been clarified in the revised manuscript. When the yield is more than 100%, it indicates that the initial concentration of NADH has been completely consumed and then got regeneration.
Page: 2, lines 77-80
“Besides, the yield of methanol was calculated based on the initial concentration of NADH, and the used equation is: Ymethanol (%) = 3*Cmethanol/CNADH, initial. In the equation, CNADH, initial is the initial NADH concentration (mM), and Cmethanol was the methanol concentration.”